# Luminance effects on pupil dilation in speech-in-noise recognition

**Yue Zhang** [1,2,3]*, **Florian Malaval**[1], **Alexandre Lehmann**[1,2,3], **Mickael L. D. Deroche**[1,2,3,4]

**1** Department of Otolaryngology, McGill University, Montreal, Canada, **2** Centre for Research on Brain, Language and Music, Montreal, Canada, **3** Centre for Interdisciplinary Research in Music Media and Technology, Montreal, Canada, **4** Department of Psychology, Concordia University, Montreal, Canada

☯ These authors contributed equally to this work.

* yue.zhang7@mail.mcgill.ca

## Abstract

There is an increasing interest in the field of audiology and speech communication to measure the effort that it takes to listen in noisy environments, with obvious implications for populations suffering from hearing loss. Pupillometry offers one avenue to make progress in this enterprise but important methodological questions remain to be addressed before such tools can serve practical applications. Typically, cocktail-party situations may occur in less-than-ideal lighting conditions, e.g. a pub or a restaurant, and it is unclear how robust pupil dynamics are to luminance changes. In this study, we first used a well-known paradigm where sentences were presented at different signal-to-noise ratios (SNR), all conducive of good intelligibility. This enabled us to replicate findings, e.g. a larger and later peak pupil dilation (PPD) at adverse SNR, or when the sentences were misunderstood, and to investigate the dependency of the PPD on sentence duration. A second experiment reiterated two of the SNR levels, 0 and +14 dB, but measured at 0, 75, and 220 lux. The results showed that the impact of luminance on the SNR effect was non-monotonic (sub-optimal in darkness or in bright light), and as such, there is no trivial way to derive pupillary metrics that are robust to differences in background light, posing considerable constraints for applications of pupillometry in daily life. Our findings raise an under-examined but crucial issue when designing and understanding listening effort studies using pupillometry, and offer important insights to future clinical application of pupillometry across sites.

## Introduction

Within hearing research, pupillometry has been shown to be a valid tool for quantifying listening effort in different listening conditions, such as with different masking noise, spectral degradation, speech intelligibility level and syntactic complexity [1–4]. Typically, when a speech recognition task gets difficult, listeners show a greater task-evoked pupillary response, until the task is so challenging that listeners 'give up'. For instance, one of the most investigated factors on listening effort is the type and level of masking noise [5–10]. Ohlenforst et al., [8] examined peak pupil dilation (PPD) across a wide range of SNR (-20 dB to + 16 dB) in

**Funding:** This research was supported by a grant from the Quebec governmet (Mitacs Accelerate

https://www.mitacs.ca/en/programs/accelerate) in collaboration with an industrial partner Oticon Medical Canada (https://www.oticonmedical.com/) [grant number IT10517]. The funding was issued to Dr. Alexandre Lehmann and Dr. Mickael Deroche, for the postdoctoral work of Dr. Yue Zhang. The funders had no role in study design, data collection and analysis, decision to publish, or preparation of the manuscript.

**Competing interests:** The authors have declared that no competing interests exist.

stationary noise and single-talker maskers. Results showed an inverse U-shaped relation between PPD and masking noise: as SNR decreased, listeners exhibited a bigger PPD until the task was so difficult that PPD was reduced again at the most adverse SNR. This highlights the non-monotonic nature of the pupil dynamics as a function of task difficulty.

While pupil dilation serves as a robust 'reporter variable' for listening effort, it is also sensitive to many other factors, among which luminosity variation is the most prominent [11,12]. The pupil size is controlled by two antagonistic smooth muscle groups, the iris sphincter and dilator muscles. When light falls on the retina, an increased neural activity in the pretectal regions and stimulation of the Edinger-Westphal nucleus leads to activation of preganglionic parasympathetic neurons and innervation of the ciliary ganglion [13,14]. These, in turn, command the constrictor muscles to tighten and lead to pupil constriction. Under the direct control of the autonomic nervous system (ANS), the pupillary response to light reflects the balance between the Sympathetic Nervous System (SNS) and the Parasympathetic Nervous System (PNS). While the range of pupillary movement in response to luminance levels can vary from less than 1mm to more than 9mm, in comparison, the largest of cognitively driven movements are about 0.5mm [1,15]. This difference in the pupillary response demonstrates that the light reaction has a much larger effect on the pupil size dynamic range than the cognitive pupillary component [16,17]. Therefore, it is important to identify and disentangle the impact of light on the pupillary response during a cognitive task, in order to validate pupillary response as a robust index for cognitive effort in ecological and likely more complex environments [18,19].

Past studies have indicated that not only is the light-induced response larger than the cognitive modulation of the pupillary response overall, but there is also an interaction between the two. However, past studies suggested inconclusive, sometimes contradictory, results. For instance, Steinhauer et al., [20] conducted two arithmetic tasks (continuously subtracting a random number by 7, i.e. difficult condition; or adding by 1, i.e. easy condition) in either dark or moderate room light. During the baseline period, no interaction was found between task difficulty and light condition. But in the response period, there was a significant interaction: the two tasks (difficulty levels) did not differ in darkness but did in moderate light (greater PPD when more difficult). Thus, pupillary changes observed during their cognitive task decreased in dark lighting conditions. Peysakhovich et al., [17] conducted a short-term memory task where participants were asked to either recall or not recall a series of auditorily presented digits. Task difficulty was controlled by the number of digits (5, 7 or 9 digits). Screen luminance changed from trial to trial among black, gray, or white. No interaction between task difficulty and light condition was observed for the baseline period, but in darker conditions, a given memory load induced higher PPD. Thus, pupillary changes observed during their cognitive task were increased by dark lighting conditions. Peysakhovich et al., [21] required participants to perform a N-back recall task coupled with an arithmetic task. Participants either added or subtracted two numbers displayed on the screen and had to respond whether the number matched the result from one block back or two blocks back. The screen was either gray (low light) or white (high light). This time, no effect of light and no interaction with task difficulty were found on the PPD, but differences in baseline pupil diameter (pre-task) between the 1-back and 2-back tasks were observed, and they were larger in low light. Larger effects of cognitive arousal on pupil size in low range of luminances was also reported in Pan et al., [22]. Participants performed auditory math problems that were either Easy or Hard continuously (4s for listening to the question and 2s for keyboard response), while different luminance levels cycled on the screen (60s for each level presentation) in each task block. The Hard condition produced larger mean pupil diameters than the Easy condition across luminance levels. However, the differences between Hard and Easy condition were larger at low- and mid-

luminances. Based on their results, authors recommended mid-luminances for pupillometry studies investigating cognitive event evoked pupillary response. Książek et al., [23] applied several analysis methods (single-value measures in the time and frequency domain, pupil time course analysis) on two data sets (data set A investigating the impact of SNR and data set B [24] investigating the impact of luminance). Results showed a significant effect of luminance in all investigated pupillary measures. This study however did not address the possibility of an interaction between SNR and luminance given the fixed level of performance in dataset B. Typically, studies investigating listening effort manipulate task difficulty levels (i.e., SNR levels, background noise types, hearing aid/cochlear implant features turned on/off, SRT levels etc.) to observe changes in pupil responses. While tasks like sentence recognition and arithmetic tasks arguably require transient investment of cognitive resources, tasks like digits recall and matching require constant straining of mental effort, and more complex tasks such as speech communication in ecological rooms require concurrent and sustained effort. Different tasks require different types of resources. As shown in previous studies, pupillary responses to concurrent cognitive tasks showed different patterns compared to single cognitive task, therefore, it is reasonable to assume that interaction between complex cognitive demands and luminance will be even more complicated [18,23,25]. Finally, to add yet another level of complexity, many findings on this very question observed within the normal hearing (NH) population may not be directly applicable to the hearing-impaired (HI) community. For example, Wang et al., [24] measured SRTs for NH and HI participants and showed that participants with better hearing acuity showed a larger difference in PPD between dark and light conditions. Participants in Pan et al. [2022] (age 18–35; no precise checks on hearing status for the auditory math task) showed heterogeneity in the luminance at which the biggest pupillary difference between Easy and Hard condition occurred. The causes of this heterogeneity are yet to be systematically examined (i.e., hearing status, age etc.). In summary, how luminance affects pupillary response during speech communication is still under-investigated, and this knowledge is important for the validity of pupillometry in clinical settings where different clinics might conduct pupillometry in different luminance levels and patients vary greatly in their pupillary dynamics.

The primary aim of the current study was to examine the impact of light level on the pupillary response using a well-replicated paradigm in listening effort research that varied task difficulty by manipulating the SNR during sentence recognition. To optimize this investigation, a preliminary experiment explored a range of four SNRs, 0, +7, +14 dB, and quiet condition. This served as a replication phase to ensure that the results were consistent with past listening effort studies (i.e. making sure our glasses/equipment functioned properly). It was also an opportunity to explore the robustness of different metrics or methods which might help overcome less-than-ideal lighting conditions. For example, pupillary response is traditionally examined by extracting a feature from the overall pupil trace (e.g., PPD amplitude or latency) or by fitting the pupil variation over time (e.g., growth curve analysis, generalised additive mixed modelling, etc). It is likely that both approaches could be confounded by different luminances. For instance, when PPD is calculated by subtracting or dividing the peak dilation by the baseline, past studies assumed that they were two independent components during a cognitive task [1], but this assumption is questionable given that light could have a differential effect on the baseline and peak window of the pupil dilation [21]. So, we considered different analysis methods (baseline subtraction, proportional change) to investigate whether the effect of luminance or its interaction with task difficulty would depend on the analytical approach considered. Different from Książek et al., [23], we will focus on PPD as the index of listening effort, due to its wide application in both research and clinical studies. In summary, the current study will explore the impact of luminance on pupillometry and highlight its importance for both the experimental design and analysis methods.

## Materials and methods

### Participants

Twenty-one listeners (11 women; 10 men) were recruited in Exp.1 from 18 to 49 years of age, with a mean (SD) of 27.3 (8.6) years. Thirty-one listeners were initially recruited for Exp.2, but three were eventually excluded (due to excessive blinks–section 3.2), reducing the sample size to 28 (21 women; 7 men) aged 18 to 51 years old, with a mean (SD) of 27.7 (9.6) years. A pure tone audiometry was administered to ensure that all participants had binaural thresholds at or better than 25 dB HL at 0.25, 0.5, 1, 2, 4, 8 kHz. All participants were native speakers of either French or English (the study being run always in their native language). This work received ethical approvals from McGill University Faculty of Medicine Research Ethics Board under the reference A05-B11-18B. Prior to the experiment, participants were given enough time to read the protocol and gave written informed consent for their participation, receiving $15 per experiment as compensation for their time.

### Stimuli

Speech stimuli were from the Institute of Electrical and Electronics Engineers (IEEE), they are phonetically balanced sentences [26] recorded from a male native American English speaker, and sentences from the Hearing in Noise Test (HINT) sentences recorded from a male Quebecois French speaker [27]. In all conditions except the quiet one, sentences were masked by speech-shaped noise. This noise was generated from the long-term excitation pattern of the entire material, respectively in English or French, and was always fixed at 65 dB SPL. Experiment 1 varied the SNR level, using 0, +7, +14 dB and quiet conditions. Experiment 2 varied both SNR and light levels, in which case only the 0 and +14 dB SNR were selected (based on the results of Exp.1). Changes in SNR were implemented by raising the target level (from 65 dB at 0 dB up to 79 dB at +14 dB, and back to 65 dB in quiet). Note that the reason to change the target level rather than the masker level was to prevent listeners from anticipating the difficulty of a block before hearing the target [9].

In experiment 2, three light levels were applied, by adjusting the room light level and screen luminance level together to reach close-to-0 lux, 75 lux, and 220 lux. The light level was measured with the luxometer (TES-1335) sensor positioned at the same height as participants' left eye and facing the screen, to approximate the amount of light hitting participants' eyes.

In each experiment, twenty sentences were tested for each condition, resulting in a total of 80 sentences in Exp.1 and 120 sentences in Exp.2. Within a block of 20 sentences, the order of the material remained the same, but the sequence of blocks was fully randomized, and different across participants.

### Procedure

Experiments were performed between February 2018 and February 2020 at the Center for Interdisciplinary Research in Music Media and Technology at McGill University, inside a sound-attenuated room. Participants sat on a rigid chair in the room, 2m in front of a 35-inch screen monitor and wearing an infrared binocular eyetracker (Tobii Glasses Pro2, 100 Hz sampling rate).

After a demo of the experiment procedure, participants firstly listened to five sentences (excluded from the test) at 14 dB SNR to familiarise themselves with the test and typical sentences of the speech material.

Before each block, the room and screen luminance levels were adjusted according to the randomisation of the Matlab program. The luminance levels were then fixed throughout the block, to avoid changes in light level inducing task-unrelated pupillary response. Listeners

were given at least 1 minute to adjust for the new light level and for the pupil to reach its light reflex target [28], plus the time necessary until they reported that they felt comfortable to continue. All audio stimuli were presented through a Beyer Dynamics DT 990 Pro headphone via an external soundcard (Edirol UA), calibrated at 65 dB SPL. Experiments were run in Matlab 2016b, using Psychtoolbox and custom software. In each trial, the presentation of the speech-shaped noise masker started 3s before the onset of the sentence. This was to provide time for the pupils to recover from the previous trial to avoid carry-over effect (2s after previous trial, initiated by the experimenter) and to measure pre-task baseline pupil diameter (1s). Participants were instructed to fixate on the black cross displayed at the center of the screen. After 3s, the sentence was played along with the continuous noise, and the presentation of the masker noise was turned off 2s after the offset of the sentence, to allow the pupil to reach its peak. Upon the masker offset, participants were prompted by the black cross turning into a circle displayed at the screen center to repeat back the sentence verbally. This delayed verbal response ensured that speech motor commands of the participants did not tarnish the pupillary response corresponding to processing of the sentence perceived. Their verbal responses were scored by the experimenter based on the number of key words correctly repeated. Then the experimenter proceeded to the next trial.

## Data analysis

**Behavioral data.** To be consistent with the statistical approach of the pupil data (where traces were aggregated per block), a generalized linear mixed-effect model was fitted on listeners' performance, averaged over the 20 sentences of each block. Specifically, a logistic model provided a suitable way to control for the ceiling effect [29]. We considered a model with one fixed factor in Exp.1 (*SNR*, *as a categorical variable*) or two fixed factors in Exp.2 (*SNR* and *luminance*, *both as categorical*) and one random factor (*subject*, *also categorical*), including random intercepts only. With a single observation per block, the model could not support any further degree of complexity. Further analyses are presented in the S1 Appendix where we considered a trial-based approach that allowed us to examine a more complex model with by-subject random slopes for each fixed factor and look specifically at the effect of position within a block. Overall, they did not provide any further insights into the behavioral data. Chi-squared tests were used to examine main effects and possible interactions. Differences between levels of each factor and interactions were examined with post-hoc Wald test. P values were estimated using the z distribution in the test as an approximation for the t distribution [30].

**Pupil data: Preprocessing.** The Tobii glasses possess four cameras, two for each eye, along with a camera recording what the participant looked at, including position of their gaze. Each camera had a sampling frequency of 50 Hz, interleaved by 10 ms (producing a pseudo 100-Hz sampling frequency). Sample points for each camera were first processed separately, removing blinks with a +/- 60 ms window on the edges and reconstructing the missing points with an autoregressive method (*fillgaps* function in Matlab). Then the signals from the two cameras were recombined, and any value below or above 3 standard deviation (SD) of the mean pupil was again counted as blinks and reconstructed using *fillgaps*. We made a further attempt to tag additional outliers by spotting gaze positions 3-SD away from the distribution of "valid" gazes (i.e., excluding gazes corresponding to points already reconstructed) using the Mahalanobis distance but this approach was too stringent in that it suspected data points that appeared perfectly reasonable.

To evaluate whether the quality of the recordings could change with experimental condition, we extracted the percentage of data points coded as blinks for each trial to use it as a dependent variable in repeated-measures analysis of variance (rm-ANOVA). In Exp.1, there

was no effect of SNR [F(3,60) = 1.4, p = 0.241], with an average blink of 11.7% per trial. In Exp.2, there was no effect of SNR [F(1,27) = 2.5, p = 0.122], no effect of luminance [F(2,54) = 1.0, p = 0.379], and no interaction [F(2,54) = 1.5, p = 0.226], with an average blink of 15.2% per trial. With regard to trial exclusion, it is common to exclude any trial that exhibits over 20% of points coded as blinks, and with this criterion, it is common to exclude any subject with over 40% trials rejected (corresponding to 8/20 trials per block in this study). Although the percentage of blinks was on average lower than this value in both experiments, this criterion would have led to several rejections of participants in Exp.2 who provided generally decent data after deblinking. Thus, we decided to allow a more liberal criterion of 45% loss to keep as many participants as possible. A previous study also supported that 45% blink exclusion criterion would not affect group pupil results [31]. On this basis, only three participants had to be excluded from Exp.2 (blinks > 45% on average across 120 sentences, leading to >8/20 trials excluded). Another rm-ANOVA was conducted on the number of excluded trials as dependent variable. In Exp.1, there was no effect of SNR [F(3,60) = 1.1, p = 0.371], with an average of 0.7/20 trials excluded. In Exp.2, there was no effect of SNR [F(1,27) = 2.5, p = 0.123], no effect of luminance [F(2,54) = 0.2, p = 0.795], and no interaction [F(2,54)<0.1, p = 0.976], with an average of 1.0/20 trial excluded.

Finally, all valid traces were low-pass filtered at 10 Hz with a first order Butterworth filter to preserve only cognitively related pupil size modulation [32], and downsampled to 50 Hz to reflect the true sampling rate of the glasses. Processed data were then aggregated per listener by SNR and luminance conditions, aligned by the onset of the response prompt (except in Additional Analyses where traces were also aligned at the sentence onset to illustrate the impact of analytical window on the results).

**Pupil data analysis.** Baseline pupil diameter in each trial was calculated as averaged pupil trace 1s before the sentence onset. The pupil diameter measured from the sentence onset to the repeat prompt was subtracted from that baseline level to obtain relative pupil diameter changes elicited by the task. In Additional analyses, to examine whether the impact of luminance was independent of the PPD baseline-correction method, we attempted a posteriori two alternative ways of calculating PPD, as a percentage of the baseline, or as a percentage of the dynamic range. In all these methods, the search for PPD was restricted to a window starting from sentence onset and ending at the verbal response prompt (thereby excluding any confound with the pupil's arousal induced by verbally repeating the sentence). Note that all sentences were left unprocessed in duration to avoid unnatural acoustic manipulation (compression or stretching of original sentences). This procedure was common in listening effort studies due to varied length of standardized sentences and the variability in sentence duration could be well controlled by consistent trace alignment (either by sentence onset or offset) [33]. Additional analysis took into account of this variability in sentence duration and alignment method and explored whether they affected our results.

A linear mixed-effect model (LME] was fitted on three dependent variables successively: baseline, PPD amplitude, and PPD latency. Each of these metrics was extracted on the trace averaged over a block (20 sentences), so there was only one observation per condition limiting the model complexity to random intercepts at most. Thus, just like for the behavioral data, we considered a model with one fixed factor in Exp.1 (*SNR, as categorical variable*) or two fixed factors in Exp.2 (*SNR* and *luminance*, *both as categorical*) and random intercepts by *subject*. Further analyses are presented (S1 Appendix) where we considered a trial-based approach that allowed us to examine more complex models with by-subject random slopes for each fixed factor and look specifically at the effect of position within a block. Chi-squared tests were used to examine main effects and possible interactions.

## Results and discussion

### Experiment 1: Effect of SNR

**Behavioral results.** As expected, speech intelligibility deteriorated as SNR decreased (Fig 1, left: SNR0, mean 89%, SD 2.3%; SNR7, mean 97.3%, SD 0.9%; SNR14, mean 98.4%, SD 0.5%; Quiet, mean 97.8%, SD 0.7%). The LME analysis confirmed a main effect of SNR [$\chi^2(3)$ = 12.6, p = 0.005]. Intelligibility was worse at 0 dB than at any other SNR (p<0.020). Importantly, this impairment was relatively minor as intelligibility still approached 90% at 0 dB (and was largely at ceiling at +7 dB and beyond). Thus, the behavioral data confirmed that we chose a range of SNRs where listeners' performance carried little information about the difficulty of the sentence recognition task. Typically, the behavioral data could not distinguish changes in task difficulty between +7 dB, +14 dB SNR, and quiet conditions.

**Pupil results.** The top panels of Fig 2 show the time course of the pupil diameter expressed in absolute unit (mm). As expected, these averaged traces exhibited a peak shortly after the end of sentences within the 2-sec window that we left before listeners were prompted to repeat the sentence. Past this prompt (time>0s in the abscissa), the pupil diameter increased again due to the verbal response and was therefore less relevant to the cognitive processes engaged in sentence decoding. There were differences in averaged pupil diameter across the four conditions, up to 0.2 mm between 0 and + 7 dB SNR. More importantly (for this type of non-demanding task), the baseline-corrected traces emphasize that the pupil dilated more strongly at 0 dB than in any other condition (bottom-left).

To substantiate these claims, a LME analysis was conducted for each of the three metrics. For *baseline*, there was no main effect of SNR [$\chi^2(3)$ = 6.2, p = 0.100], suggesting that the small differences aforementioned (Fig 2, top-right) were not meaningful. For *PPD amplitude*, there was a main effect of SNR [$\chi^2(3)$ = 22.8, p<0.001] driven by larger PPDs at 0 dB than at any other SNR (p<0.001). Estimates were -0.07, -0.09, and -0.08 mm, respectively for 7 dB, 14 dB, and quiet conditions relative to the 0 dB condition (Fig 2, bottom-middle; SNR0, mean 0.21mm, SD 0.03mm; SNR7, mean 0.14mm, SD 0.03mm; SNR14, mean 0.12mm, SD 0.03mm; quiet, mean 0.13mm, SD 0.02mm). For *PPD latency*, there was no main effect of SNR [$\chi^2(3)$ =

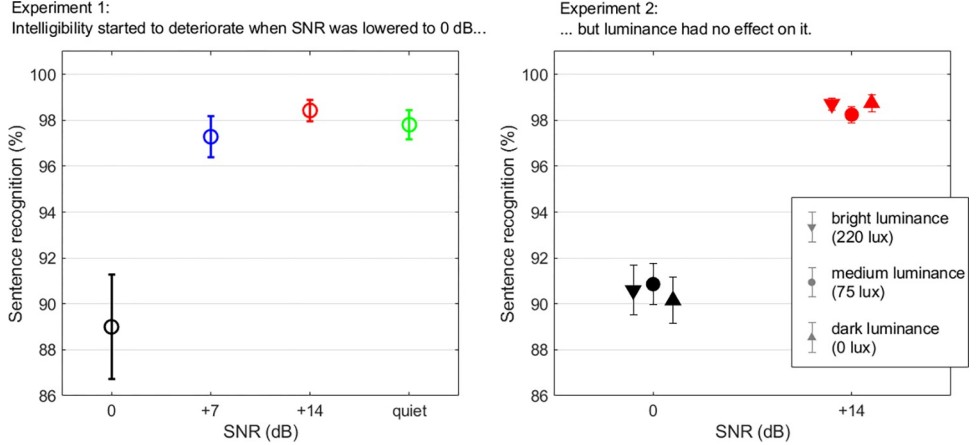

**Fig 1. Performance in the sentence recognition task.** Lists of twenty sentences presented at different SNRs in Exp.1 (left panel), in which 21 listeners sat in a normally-lit room (75 lux). Exp.2 measured performance in the same task (using different sentence lists than in Exp.1) at two SNRs in which 28 listeners sat in the same room under similar luminance settings (75 lux), or in close-to-darkness (0 lux), or in very bright luminance settings (220 lux). Luminance (although it has a huge impact on the pupil) had no effect on behavioral performance. Error bar represents 1 standard error from the mean.

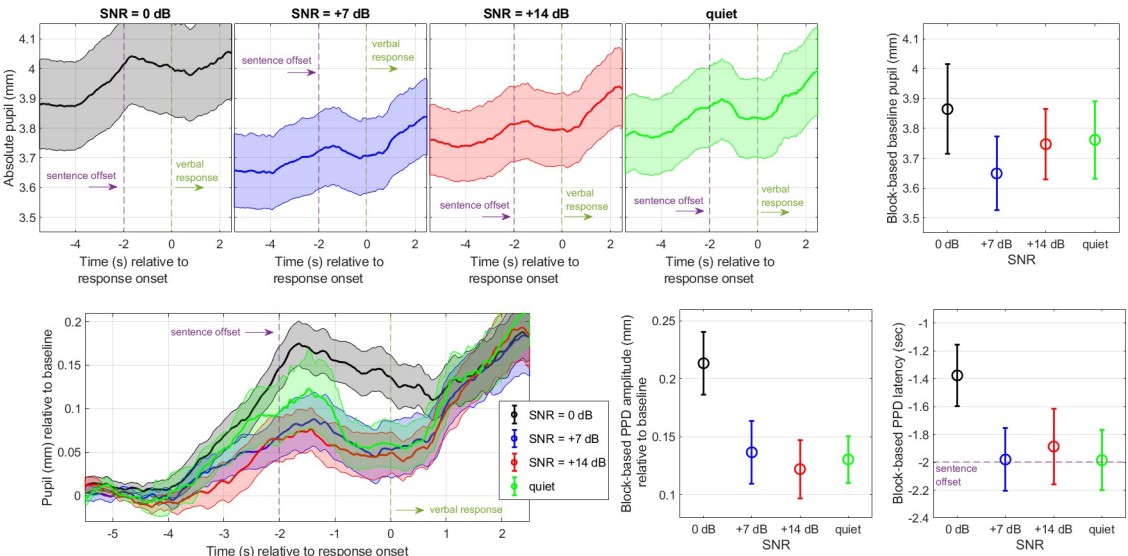

**Fig 2. Pupil responses in Experiment 1.** Mean pupil traces across four SNR conditions, measured for 21 listeners in Experiment 1. Traces are expressed in absolute unit (top) or relative to the 1-sec baseline (bottom). Three metrics were extracted: Baseline (top-right), PPD amplitude (bottom-middle), and PPD latency (bottom-right). The pupil dilated more at 0 dB than any other SNR. Error bar and shaded width represents 1 standard error from the mean.

5.8, p = 0.122]. Despite the pattern depicted (Fig 2, bottom-right), differences across SNRs were too weak to be significant, with this block-based approach. Note that a trial-based approach (S1 Appendix) appeared more sensitive to latency differences and did reveal a later PPD at 0 dB than at any other SNR.

**Discussion.** Exp1 replicated the effect of SNR on pupillary response [8,9], suggesting that our experimental setups and devices were sensitive and valid enough to capture task-evoked pupillary responses. A SNR of 0 dB elicited significantly bigger PPD and longer latency than +7, +14 dB SNR and quiet conditions, even when sentence intelligibility was at ceiling. The choice of SNR in our study was easier than in previous studies, which explains the ceiling in sentence recognition and lack of significant difference between 7 and 14 dB SNR in PPD and latency. This SNR choice is however very relevant to cochlear implant (CI) users (our future research) who generally need positive SNRs to understand speech above 50%. Despite this SNR choice being easier than most studies on NH listeners, our results still show consistent pattern with the literature. We also replicated a widely reported finding that PPD is larger when sentences are misunderstood or incorrectly repeated [44] (see S2 Appendix). Therefore, we proceeded to Exp2 by selecting 0 dB and 14 dB SNR due to the significant difference observed between the two conditions (and quiet being a qualitatively different setting).

## Experiment 2: Effect of luminance

**Behavioral results.** Speech intelligibility deteriorated as SNR decreased from +14 to 0 dB, replicating the performance levels obtained in Exp.1 (SNR0, mean 90.5%, SD 1%; SNR14, mean 98.6%, SD 0.3%). Furthermore, intelligibility did not change depending on whether participants listened in a very bright or dark room (Fig 1, right: bright, mean 94.7%, SD 0.7%; medium, mean 94.6%, SD 0.6%; dark, mean 94.5%, SD 0.7%). The LME analysis revealed a main effect of SNR [$\chi^2(1) = 158.4$, p<0.001], but no main effect of luminance [$\chi^2(2) = 0.2$, p = 0.917] or of its interaction with SNR [$\chi^2(2) = 0.8$, p = 0.679]. Therefore, listeners'

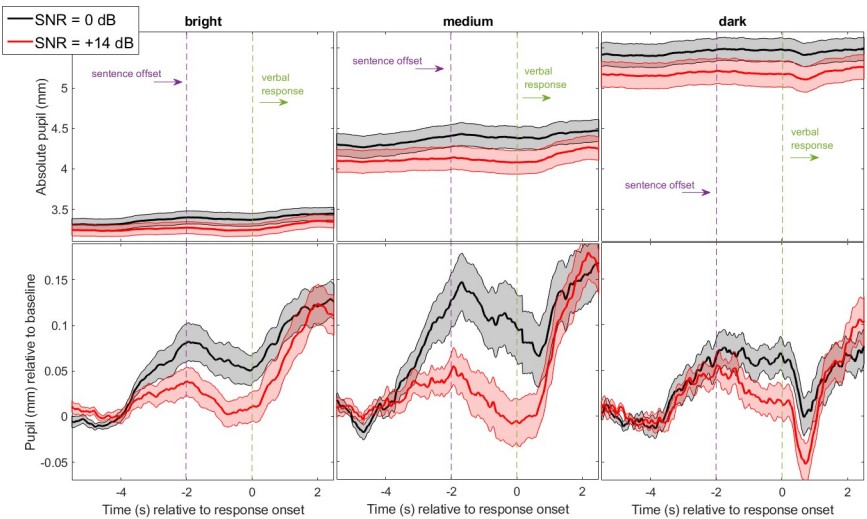

**Fig 3. Pupil responses in Experiment 2.** Mean pupil traces recorded for 28 listeners in Experiment 2: a sentence intelligibility task at two difficulty levels (0 and +14 dB SNR) under three luminance settings (dark, medium, and bright). Traces are all aligned by the onset of verbal response, and expressed in absolute units (top panels) to illustrate the massive light-evoked differences in pupil's diameter, or corrected by their 1-sec baseline (bottom panels) to emphasize the peak pupil dilation. Shaded width represents 1 standard error from the mean.

performance was the same across the three luminance settings, and (being at 90% and above) it carried little information about the task difficulty.

**Pupil results.** Visual inspection of Fig 3 suggested that three luminance settings induced big differences in pupil diameter across all subjects, going as low as 2.7 mm on average in bright luminance and as high as 6.8 mm in darkness. In addition, the pupil appeared bigger at 0 dB than at +14 dB, in line with our expectations. Once again, these averaged traces exhibited a peak shortly after the end of sentences, which was easier to observe with baseline-corrected traces (Fig 3, bottom). The PPD induced by the easy task (+14 dB SNR) was roughly similar across the three luminance settings, and of small magnitude. In contrast, the PPD induced by the (slightly more) difficult task was overall larger but it was less pronounced in either darkness or brightness as compared to that observed in medium luminance.

To substantiate these claims, LME analyses were conducted for each of the three pupil metrics.

*Baseline*: There was no main effect of SNR [$\chi^2(1) = 1.2$, p = 0.263], but a main effect of luminance [$\chi^2(2) = 225.7$, p<0.001] without interaction [$\chi^2(2) = 1.3$, p = 0.529]. The pupil baseline reacted hugely to luminance (Fig 4, left) but as in Exp 1 there was not sufficient evidence that it depended on SNR.

*PPD amplitude*: There were both a main effect of SNR confirming larger PPDs at 0 than at +14 dB [SNR0, mean 0.15mm, SD 0.02mm; SNR14, mean 0.1mm, SD 0.01mm; $\chi^2(1) = 11.8$, p<0.001] and a main effect of luminance [bright, mean 0.1mm, SD 0.02mm; medium, mean 0.16mm, SD 0.03mm; dark, mean 0.13mm, SD 0.02mm; $\chi^2(2) = 11.8$, p = 0.003] without interaction between the two [$\chi^2(2) = 2.4$, p = 0.304]. PPDs were larger under medium luminance than in brightness (p = 0.002) or in darkness (although this latter comparison was not significant, p = 0.074) (Fig 4, middle). In other words, although the traces hardly differentiated the two SNRs in either bright or dark luminance, the current evidence is that they were both suboptimal at 0 or 14 dB, compared to medium luminance settings. This is one of the key findings of this article.

Extreme luminance settings (0 or 220 lux) were both suboptimal in capturing the PPD, relative to normal luminance (75 lux), but had little effect on PPD latency. They had however a huge impact on average diameter, or baseline, with an effect size 10-20 times greater than the average PPD.

**Fig 4. Pupil metrics in Experiment 2.** Three pupil metrics, namely baseline (left), PPD amplitude (middle) and latency (right), extracted from block-averaged pupil traces recorded in experiment 2. Error bar represents 1 standard error from the mean.

*PPD latency*: There was a main effect of SNR confirming later PPDs at 0 than at +14 dB [SNR0, mean -0.5s, SD 0.2s; SNR14, mean -0.04s, SD 0.3s; $\chi^2(1)$ = 10.8, p = 0.006], an effect that was revealed in Exp.1 only with the trial-based approach (S1 Appendix) but not with the block-based approach, suggesting it was presumably a power limitation. There was no effect of luminance [$\chi^2(2)$ = 3.4, p = 0.185] or interaction [$\chi^2(2)$ = 1.0, p = 0.606]. On average across luminance settings, the PPD occurred 432ms later at 0 dB relative to 14 dB (Fig 4, right).

**Discussion.** Exp2 showed that the magnitude and timing of pupillary responses varied in different light settings, suggesting that task-evoked pupillary responses were affected by the luminance level. For baseline, luminance affected the absolute pupil diameter evenly across SNR conditions. Arguably, for a sentence recognition task without cognitive involvement prior to the trial (i.e., passively listening to the background noise), the response of pupil diameter in different luminance is dominated by the ANS [12]. As a reminder, SNS directly controls the pupil dilator muscle and PNS indirectly controls the pupil sphincter (constrictor) muscle. Pupillary light reflex is regulated by the PNS pathway [14,34], and when the cognitive stage kicks in (i.e., listening to a target sentence and preparing for verbal repetitions), pupillary dilation induced by task difficulty is regulated by PNS inhibition and SNS activity [20,35]. Firstly, the order of magnitude is remarkable: PPDs (of about 0.1 to 0.2 mm) were on the order of 10–20 times smaller in magnitude than luminance-evoked changes, consistent with previous studies on the comparative effect size of luminance-evoked and task-evoked pupillary responses [1,15–17]. Secondly, although the task difficulties were the same in each luminance condition (and the behavior results supported this idea), task-evoked PPDs were different. For a given SNR level, when measured in dark (0 lux) and bright (220 lux) environments, the PPDs were smaller than in medium luminance (75 lux) environment. It seems that both extreme constriction and dilation restrain the range of task-evoked pupillary response, even after baseline correction. In other words, in those sub-optimal luminance conditions, it is likely that PPD is underestimated. The latency of PPDs, on the other hand, seems relatively unaffected by luminance (see Additional analysis on how the calculation method of latency could change this finding). Our finding that medium luminance is more conducive of larger task-evoked pupillary response than bright luminance is consistent with some previous findings [21,22]. But decreasing the luminance from medium to dark did not further increase the pupillary response as in Pan et al. [22], instead, in our current study, PPD in dark tended to be smaller than in medium luminance. The difference could be due to the calculation method of pupillary responses: in Pan et al. [22], pupillary responses were calculated as the mean pupil size which contained both the baseline and task-evoked response; in our current study, PPD was baseline-

corrected to better capture time-aligned pupillary response to sentence recognition. It is likely that after correcting for the baseline which increased hugely in the dark, the benefit of low-luminance in inhibiting PNS when calculating PPD is scaled correspondingly. However, this scaling has been suggested to be linear, as demonstrated in Reilly et al., [36], therefore, PPD should not decrease from medium luminance to dark if the only factor that had changed is the decreased contribution of PNS. Perhaps, individual differences could introduce some non-linear factors in the pupillary response. Some evidence for this possibility can be glimpsed from Pan et al., [22], who reported low group agreement specifically in low luminances: while all participants showed consistent pattern at medium luminances, there were a few participants who showed no modulation at low luminances. But participants in our study and in Pan et al. [22] were relatively homogenous, leaving little room for investigating person-specific factors. Note that identifying and mapping out linear or non-linear relationships across participants on pupillary measures will not only help to control for confounds in task-evoked interpretations, but also provide meaningful methods to investigate complex interactions between PNS and SNS [20,36].

The lack of significant interaction between luminance and SNR conditions in our results show that this bias is relatively consistent across SNRs, suggesting that the luminance might not interfere greatly with experimental conditions or task difficulty in pupillary responses (e.g., a difficult task will always elicit a bigger pupillary response than an easy task, and incorrectly repeated sentences elicit bigger pupillary response than correctly repeated sentences, see S2 Appendix). However, a closer examination at the trend of our results raises potential issues. Note that from dark to bright luminance levels, PPD showed an inverted U-shape (Fig 4), suggesting that it is possible that there exists a luminance level where there will be an even bigger SNR contrast in the PPD response, and the interaction between SNR and luminance might reach significance at the 'tipping point' of that inverted U-shape (as already shown in the bigger separation of error bars in medium compared to dark/bright luminance conditions). A similar inverted U-shape was found by Pan et al. [22], where the biggest and most consistent difference in pupillary response between Hard and Easy auditory math tasks occurred at medium luminance levels among all ten luminance levels tested. To confirm this speculation, future studies need to apply a wider range of changes in the luminance to map out the entire psychometric function between luminance and task-evoked pupillary response. Without this knowledge, we might not be confident enough to synthesize knowledge across research sites to understand and compare the effect of task difficulty on pupillary response. To illustrate, for instance, even when Lab A and Lab B used the same devices, experimental designs and analytical pipelines, Lab A might observe an effect that is bigger than in Lab B, just because Lab A chose a luminance level that is closer to the 'tipping point' of the inverted U-shape.

### Additional analyses

To further understand the complexity of the additive effect of pupillary light reflex and task-evoked pupillary response, we performed additional analyses to explore whether the impact of luminance we had observed could be affected by the analysis methods. Specifically, we examined baseline correction methods when calculating PPD and the impact of sentence duration. As discussed above, applying baseline correction might be partially responsible for the differences in task-evoked pupillary responses observed in our current study and in Pan et al. [22]. Several methods have been proposed in previous studies but it is unclear whether they could control for the impact of the luminance level on task-evoked pupillary responses. Also, sentence duration varies and this variability contributes to the variability of the analytical window (sentence onset to the repeat prompt), hence possibly generating variability in PPD amplitude and latency measurement.

## Effect of PPD calculation methods

Up to now, we calculated PPD in mm by substracting the pupil diameter in the analytical window (sentence onset to the repeat prompt) by the baseline diameter. Arguably, other methods have been proposed to calculate PPD [33,36,37]. Here, we explored two alternatives, 1) as a percentage relative to baseline [38,39], and 2) as a percentage relative to the dynamic range of a given subject [40]. This latter method was particularly suited to Exp.2, where the range evoked by luminance differences was considerable. To this aim, we pulled all 120 trials for a given subject, and extracted the 0.5 and 99.5 percentile of the distribution of sample points to define the luminance-evoked dynamic range for a given subject. In Exp.1, we could not access the same metric but we followed the same approach to access the task-evoked dynamic range across the four SNRs (80 trials). Note that this is different from other approaches where the dynamic range was measured outside of the task when the subject is at rest [41]. The dynamic range used here was presumably larger than it would have been—had it been extracted prior to the experimental protocol.

Fig 5 shows the averaged traces in each experiment, for the first (top) and second alternative (bottom). A LME analysis was reiterated for these two alternative ways of calculating PPD amplitude (all results were identical for PPD latency). Expressed as a *proportional change from baseline*, there was a main effect of SNR [$\chi^2(3) = 16.6$, p<0.001] driven by PPDs between 1.7 to 2.2% larger at 0 dB than at any other SNR in Exp.1 (p<0.001) (Fig 5, top-left). In Exp.2, a considerably large baseline in darkness would underestimate the PPD amplitude whereas a small baseline in brightness would potentially overestimate it (Fig 5, top-right). A LME analysis revealed both a main effect of SNR [$\chi^2(1) = 10.3$, p = 0.001] and a main effect of luminance [$\chi^2(2) = 8.3$, p = 0.016] without interaction [$\chi^2(2) = 2.6$, p = 0.275]. This approach would likely better preserve SNR differences in bright luminance and hinder them in dark luminance, but the main findings were unchanged qualitatively. In the second alternative, expressed as a *proportional change relative to the dynamic range*, there was a main effect of SNR in Exp.1 [$\chi^2(3) = 22.3$, p<0.001] driven by PPDs between 4.7 to 5.8% larger at 0 dB than at any other SNR (p<0.001) (Fig 5, bottom-left). In Exp.2, there were both a main effect of SNR [$\chi^2(1) = 14.9$,

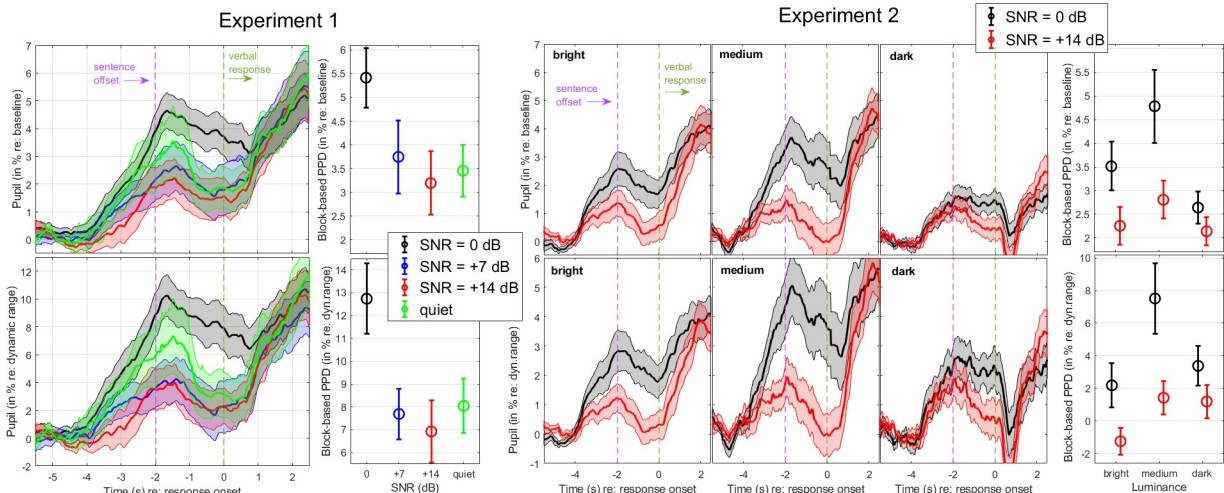

**Fig 5. Averaged pupil traces and metrics.** Averaged pupil traces and their respective block-averaged PPD amplitudes expressed as a proportional change from baseline (top) or as a proportional change relative to the dynamic range (bottom), in both experiments. In Exp.2, the dynamic range was much greater than in Exp.1 (as it was induced by luminance differences instead of subtle task differences such as SNR), resulting in a narrower scale of percentage changes. Error bar and shaded area represented 1 standard error from the mean.

p<0.001] and a main effect of luminance [$\chi^2(2) = 11.3$, p = 0.003] without interaction [$\chi^2(2) = 2.5$, p = 0.288] (Fig 5, bottom-right). Once again, the results were qualitatively unchanged. In other words, we conclude that one cannot "repair" the poor sensitivity of pupil reading in darkness or in very bright luminance by choosing a better method to calculate PPD.

**Effect of sentence duration.** The longer the sentence, the more decoding must take place, and as a result, the pupil response may increase and be delayed incrementally as a function of sentence complexity [42]. To simplify this problem, here, we only looked at their duration (i.e., making a debatable assumption that a longer sentence necessarily contained more complex content). We grouped all materials into five duration bins, with centers corresponding to 1660, 2060, 2280, 2460, and 2700 ms. These values were extracted from the 10th, 30th, 50th, 70th, and 90th percentile of the distribution of all sentence materials. Adding this variable as a fixed factor in the aforementioned LME model, we found a main effect of *duration* on PPD amplitude [$\chi^2(1) = 6.3$, p = 0.012], but no interaction with *SNR* [$\chi^2(3) = 5.2$, p = 0.155]. The main effect of *duration* did not reach significance for PPD latency [$\chi^2(1) = 2.5$, p = 0.110], and there was no interaction with *SNR* [$\chi^2(3) = 5.7$, p = 0.126]. Note that this effect of *duration* on latency was certainly present when traces were aligned relative to sentence onset [$\chi^2(1) = 14.9$, p<0.001] (the interaction with *SNR* remaining absent [$\chi^2(3) = 5.3$, p = 0.144]). This suggests that the effect of sentence duration on latency is largely taken care of by aligning the responses from the sentence offset (and is a good reason why one may want to follow this recommendation–see comparison of top and bottom panels in Fig 6). Exp.1 thus confirmed that PPD is generally larger with longer (and likely more complex) sentences, disregarding SNR. The question arose as to whether the same could be said of bright and dark luminance settings.

In Exp.2, the sentence lists used were slightly different from Exp.1, so the five duration bins had centers at 1680, 2040, 2280, 2500, and 2760 ms, once again extracted from the 10th, 30th, 50th, 70th, and 90th percentile of the respective materials. The factor *duration* did not lead to a main effect on PPD amplitude [$\chi^2(1) = 0.3$, p = 0.578], and there was no interaction with *SNR*

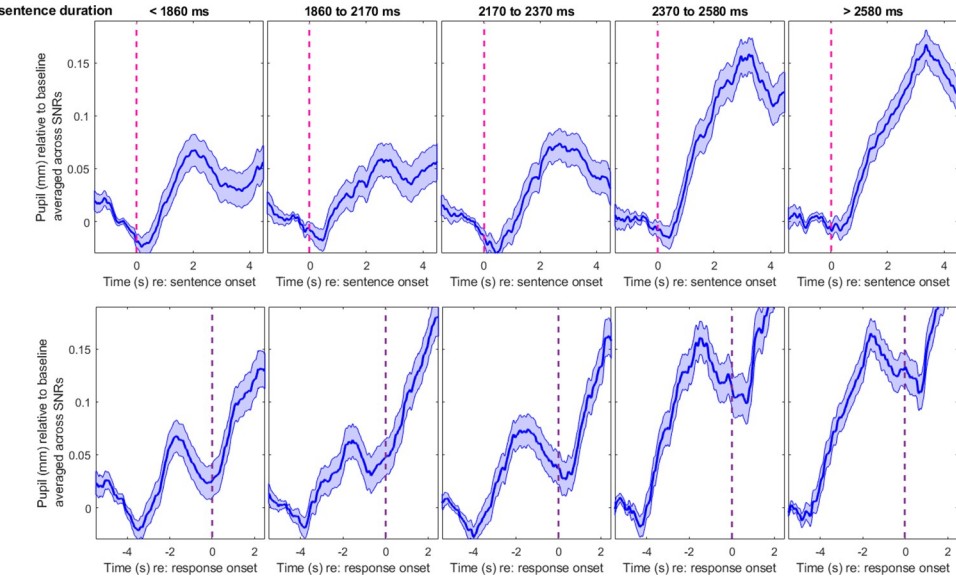

**Fig 6. Pupil metrics across different sentence durations in Experiment 1.** Baseline-corrected pupil traces measured in Exp.1 across different duration bins, aligned from the sentence onset (top) or from the response prompt (bottom). Longer sentences elicited larger PPDs, and whether this was accompanied by later PPDs depended on time alignment: The delayed PPD with longer stimuli is largely cancelled when sentences are aligned by their offset. Shaded area represented 1 standard error from the mean.

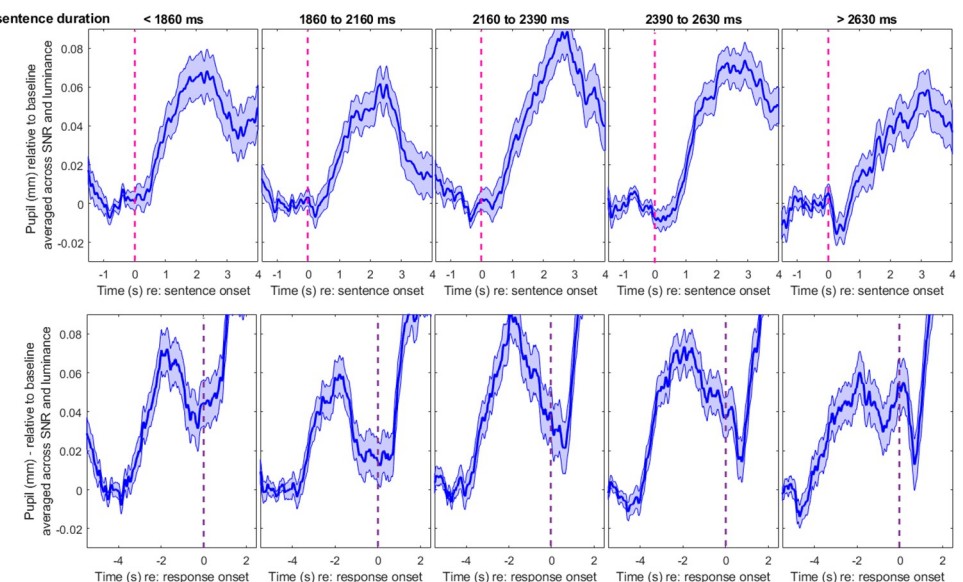

**Fig 7. Pupil metrics across different sentence durations in Experiment 2.** Baseline-corrected pupil traces measured in Exp.2 across different duration bins, aligned from the sentence onset (top) or from the response prompt (bottom). Longer sentences elicited larger PPDs, and whether this was accompanied by later PPDs depended on time alignment: The delayed PPD with longer stimuli is largely compensated for when sentences are aligned by their offset. Shaded area represented 1 standard error from the mean.

$[\chi^2(1)<0.1, p = 0.757]$, *luminance* $[\chi^2(2) = 1.3, p = 0.526]$, or in a 3-way $[\chi^2(2) = 2.3, p = 0.317]$. For PPD latency, there was a main effect of *duration* $[\chi^2(1) = 5.6, p = 0.018]$, interacting with *SNR* $[\chi^2(1) = 5.6, p = 0.018]$, but not with *luminance* $[\chi^2(2) = 1.0, p = 0.594]$, or in a 3-way $[\chi^2(2) = 0.1, p = 0.946]$. These results were not straightforward to interpret. On one hand, luminance never interacted in these analyses, suggesting that it would not play any role. On the other hand, this analysis failed to replicate larger PPDs for longer sentences and instead found dubious effects on latency. Fig 7 shows the averaged traces for each duration bin, aligned by sentence onset (top) or sentence offset exactly 2 seconds before the prompt to respond verbally (bottom). No direct relationship was apparent between the size of the PPD and the sentence duration (as it was in Fig 6). As for its latency, any effect of duration was minimal (including at each SNR separately, not plotted here). Note that the effect of *duration* on latency was present when traces were aligned from sentence onset $[\chi^2(1) = 22.3, p<0.001]$, as in Exp.1, without interaction with *SNR* $[\chi^2(1) = 3.8, p = 0.051]$, luminance $[\chi^2(2) = 0.2, p = 0.918]$, or in a 3-way $[\chi^2(2) = 0.8, p = 0.679]$. So, the PPD did occur later with longer sentences at least with respect to sentence onset, but it remains puzzling that PPD amplitude did not follow the expected trend. One explanation is that the dynamics of the pupils recorded in dark or bright luminance tended to be less stereotypical (in addition to being of smaller magnitude overall after aggregation across trials) than in medium luminance; and for this reason, they did not react to sentence duration as systematically as they would have in medium luminance. This interpretation would point to an interaction between duration and luminance, but perhaps this interaction is subtle to obtain and would require more power (than the 40 sentences per luminance level used here). The striking contrast between Figs 6 and 7 makes us conclude that dark or bright luminance settings are not only less-than-ideal in terms of PPD amplitude, they also make the traces less dependent on sentence duration.

## Conclusion

Our results raised an under-examined but crucial issue when designing studies using pupillometry, particularly in the context of speech communication. Although previous studies and Exp1 confirmed that pupillary response is a reliable measure of listening effort in simple tasks, Exp2 suggested that the luminance of the experimental setup could affect the magnitude (and possibly the significance level) of the observed task-evoked pupillary response, due to the overriding ANS impact on the pupillary muscle system. We could not map out the entire relation between luminance and PPD, due to 1) the long time required for allowing participants to adjust to many new luminance levels at different SNRs, and 2) physical constraints of our lab (i.e., luminance level could not exceed 220 lux). But the results of Exp2 suggest a main effect of luminance on PPD and possibly an inverted U-shape relation between luminance condition and PPD, which is sufficient to raise concerns on the validity and reliability of pupillometry studies: effects observed in pupillary response are likely to be confounded by the luminance level of the experimental setting. Although most pupillometry studies report luminance level, there are inconsistencies in the method (e.g. measured close to the screen, directly at the participant's eyes, or as ambient light in the room) and inconsistencies in the unit reported (lux vs cd/m2). Also, SNR manipulation was used in Exp2 because the impact of SNR on pupillary response was well validated in past studies. But other cognitive aspects and task difficulty have been examined using pupillometry, for instance working memory capacity, spectral resolution, divided attention, background noise type etc [3,43,44], and it is very likely that these effect sizes also depend on luminance level. With pupillometry being more and more popular, this fundamental question is increasingly important to address before the technique further expands into clinical applications.

Unfortunately, as suggested in Książek et al., [23] and our Additional analyses, there might not be an effective method to correct for the confound of luminance at the post-hoc analytical stage, due to the convoluted impact of ANS and central cognitive processing on the pupillary response. PPD latency was here relatively robust to luminance difference, but PPD latency is generally not as sensitive and as widely used in pupillometry studies compared to PPD amplitude. Certainly, we have not exhausted all the possible analytical measures, but this challenge further highlights the importance to understand and control for luminance at the planning and execution stages. A guideline (similar to [33,37]) might be useful for the research community to standardize the execution and report of luminance that could potentially bias the observed effect size of factors of interest. Our results add a cautious note to future clinical applications of pupillometry. Consistent with past studies and guidelines, we do recommend medium luminance based on our findings, but uniform luminance setting might not be realistic for all clinics. Therefore, to ensure the validity and comparability of pupillometry studies across clinics, a system with integrated and better luminance control might be preferable. For instance, virtual reality system with enclosed eyetracker and highly controlled visual field could be a standardised model for distribution.

To summarize, pupillometry remains a powerful tool to reveal the hidden cost of speech communication. To further apply this tool in clinical settings, we need strict examinations of factors that can affect task-evoked pupillary response, in order to enhance the validity and generalizability of pupillometry in cognitive hearing and clinical research.

## Supporting information

**S1 Appendix. Results using trial-based approach for calculating PPD.**
(DOCX)

**S2 Appendix. Comparing pupil responses for correctly and incorrectly repeated sentences.** (DOCX)

**S1 File.**
(7Z)

## Acknowledgments

We are grateful to all participants for their time and effort. We are also grateful to the constructive feedback from our reviewers and academic editor.

## Author Contributions

**Conceptualization:** Yue Zhang, Alexandre Lehmann, Mickael L. D. Deroche.

**Data curation:** Yue Zhang, Florian Malaval.

**Formal analysis:** Yue Zhang, Mickael L. D. Deroche.

**Funding acquisition:** Alexandre Lehmann, Mickael L. D. Deroche.

**Investigation:** Yue Zhang, Alexandre Lehmann, Mickael L. D. Deroche.

**Methodology:** Yue Zhang, Florian Malaval, Alexandre Lehmann, Mickael L. D. Deroche.

**Project administration:** Yue Zhang, Alexandre Lehmann, Mickael L. D. Deroche.

**Resources:** Alexandre Lehmann, Mickael L. D. Deroche.

**Software:** Yue Zhang, Mickael L. D. Deroche.

**Supervision:** Alexandre Lehmann, Mickael L. D. Deroche.

**Validation:** Yue Zhang, Mickael L. D. Deroche.

**Visualization:** Yue Zhang, Mickael L. D. Deroche.

**Writing – original draft:** Yue Zhang, Mickael L. D. Deroche.

**Writing – review & editing:** Yue Zhang, Mickael L. D. Deroche.

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
