## [Decision Letter · Decision Letter 0]

18 May 2022

PONE-D-22-07841Luminance effects on pupil dilation in speech-in-noise recognitionPLOS ONE

Dear Dr. Zhang, Thank you for submitting your manuscript to PLOS ONE. I have received two reviews; one reviewer prefers to remain anonymous; the other is Jamie Reilly. I have also read the manuscript myself. As you will see, the reviewers are generally positive about your work, but feel that there should be a more thorough discussion of previous work on the same topic. I agree with their assessment. Right now, you focus mostly on Steinhauer's 2004 paper, but this line of research actually goes back to the 60s, and both reviewers provide several relevant references. Therefore, I invite you to address this concern, as well as the other points raised by the reviewers, in a revision.

We look forward to receiving your revised manuscript.

Kind regards,

Sebastiaan Mathôt, Ph.D.

Academic Editor

PLOS ONE

Journal Requirements:

2. Please change "female” or "male" to "woman” or "man"" as appropriate, when used as a noun (see for instance https://apastyle.apa.org/style-grammar-guidelines/bias-free-language/gender).

“This research was supported by a grant from the Quebec governmet (Mitacs Accelerate) in 607 collaboration with an industrial partner Oticon Medical Canada [grant number IT10517]. We are 608 grateful to all participants for their time and effort.”

“This research was supported by a grant from the Quebec governmet (Mitacs Accelerate https://www.mitacs.ca/en/programs/accelerate) in collaboration with an industrial partner Oticon Medical Canada (https://www.oticonmedical.com/) [grant number IT10517]. The funding was issued to Dr. Alexandre Lehmann and Dr. Mickael Deroche, for the postdoctoral work of Dr. Yue Zhang.

The funders had no role in study design, data collection and analysis, decision to publish, or preparation of the manuscript”

5. Please upload a new copy of Figure 2, 4, 3, 5, 6, 7 and 8 as the detail is not clear. Please follow the link for more information: https://blogs.plos.org/plos/2019/06/looking-good-tips-for-creating-your-plos-figures-graphics/

Reviewers' comments:

Reviewer's Responses to Questions

**Comments to the Author**

1. Is the manuscript technically sound, and do the data support the conclusions?

Reviewer #1: Yes

Reviewer #2: Yes

2. Has the statistical analysis been performed appropriately and rigorously? 

Reviewer #1: Yes

Reviewer #2: Yes

3. Have the authors made all data underlying the findings in their manuscript fully available?

Reviewer #1: Yes

Reviewer #2: Yes

4. Is the manuscript presented in an intelligible fashion and written in standard English?

Reviewer #1: No

Reviewer #2: Yes

5. Review Comments to the Author

Reviewer #1: This manuscript describes a study to the effect of luminance and SNR on the peak pupil response dilation. Although the study methods are sound, the embedding of the study in the context of existing study can be improved substantially. Please find my suggestions below.

1. Embedding in existing literature. The authors indicate that the study described in the manuscript is one of few studies addressing the combined influence of luminance and (auditory) task load on the pupil response (e.g. line 562). However, the authors did not include a rigorous description of current work. Specifically, the authors should include the studies cited below (and probably more relevant studies can easily be found) to provide a more comprehensive overview of the current state of knowledge. The current studies up to date suggest that the current results are inconclusive, namely some studies have found evidence for an effect of luminance on the PPD evoked by cognitive tasks, and some did not show this effect. The authors could expand the discussion about the factors that may be associated with this current inconclusive evidence.

2. Currently, most published “guidelines” on pupillometry recommend to use a medium illumination level. How would the current results affect this recommendation? Importantly, the authors report a main effect of luminance, and the post-hoc analysis shows that the pupil response in bright conditions differs from that in medium level conditions. However, the difference between the dark and medium level condition is not significant. Regardless, the remaining part of the manuscript treats this absent effect as if this difference was indeed significant (e.g. line 395). Please adapt (why would you test the main effect if you don’t follow up on the results appropriately?).

3. Related to the incomplete account of the current literature, the authors do not explain and discuss the potential physiological mechanisms that may account for (part of) the effects. Specifically, although Steinhauer’s work is cited, they do not discuss the notion that the pupil response in darkness might be affected by a limited effect of the PNS on the pupil size in dark. Please include a discussion on the separate roles of PNS and SNS in the pupil response (and the effect of illumination)

4. The authors suggest that luminance effects on the pupil response to cognitive load may be a negative aspect, and that it may “hinder” (e.g., line 82, line 120) the pupillometry applications. Rather, such effects can be exploited in case these are meaningful (e.g. Steinhauers work). Please discuss the potential benefits of such approaches.

5. Line 68: a larger effect does not automatically indicate a predominant effect. Please rephrase.

6. Line 136 and line 386: treating the baseline as being associated with the “tonic LC state” is probably not appropriate in the current study, as the baseline was derived just before the actual sentence presentation. The tonic LC state is observed during rest. The participants are not passively listening but probably actively anticipating the target speech. See also Joshi, S., & Gold, J. I. (2020). Pupil size as a window on neural substrates of cognition. Trends in cognitive sciences, 24(6), 466-480.

7. Please provide the rationale and hypotheses for the (additional) analyses performed. Please also provide the rationale for the specific analyses performed on the behavioural data. Are these analyses robust against the fact that the behavioural performances were at ceiling levels?

8. The sentence duration needs to be provided in the main methods section. How is the varying sentence length accounted for with respect to the interval in which the pupil data were analysed? Was the interval adapted relative to the length of the shortest sentence? The additional analyses do not support this.

9. In many studies, the dynamic range of the pupil size that is used in the proportional baseline correction is specifically and separately measured (not during the actual task). This differs from the current approach. Could this have affected the results?

10. The resolution of the figures is poor, also when downloading the images directly from the portal.

11. The authors already interpret the data in the results section (e.g. lines 374-375). In addition, the authors could use more formal language throughout the paper (e.g. line 566: what is meant with “gripping”?).

12. Line 409: serperation typo

13. Lines 407-409: this is speculative; please indicate that this is not based on the current evidence.

Pan, J., Klímová, M., McGuire, J.T. et al. Arousal-based pupil modulation is dictated by luminance. Sci Rep 12, 1390 (2022). https://doi.org/10.1038/s41598-022-05280-1

Van der Stoep, N., Van der Smagt, M.J., Notaro, C. et al. The additive nature of the human multisensory evoked pupil response. Sci Rep 11, 707 (2021). https://doi.org/10.1038/s41598-020-80286-1

Madsen, J., Julio, S. U., Gucik, P. J., Steinberg, R., & Parra, L. C. (2021). Synchronized eye movements predict test scores in online video education. Proceedings of the National Academy of Sciences, 118(5).

Reilly, J., Kelly, A., Kim, S. H., Jett, S., & Zuckerman, B. (2019). The human task-evoked pupillary response function is linear: Implications for baseline response scaling in pupillometry. Behavior research methods, 51(2), 865-878.

Reviewer #2: This well-written and interesting work highlights a question of methodological significance in pupillometry (i.e., how should we account for variable lighting conditions?). The study was conducted with admirable rigor, and the data will contribute to our evolving understanding of how to optimize pupillometry and control for the all-important factor of luminance. This reviewer had several questions that the authors might wish to consider prior to this work:

One of the central claims of this work is that little is known about the interaction between task difficulty and luminance that might inform best practices in baseline correction. I would argue on this point that this is not entirely correct. The authors have missed several studies examining the effects of luminance perturbations on the magnitude of the task-evoked pupillary response function (Pan et al., 2022; Reilly et al., 2018). In addition, there are other past studies uncited (e.g., Bradhshaw, 1969) that involved similar manipulations from a half century ago. It is reasonably well-accepted that linear (subtraction) is the most biologically plausible form of baseline pupil correction (as referenced by Mathot et al). It might be useful to make contact with some of the following sources on the effects of luminance:

Bradshaw, J. L. (1969). Background light intensity and the pupillary response in a reaction time task. Psychonomic Science, 14(6), 271–272. https://doi.org/10.3758/BF03329118

Pan, J., Klímová, M., McGuire, J. T., & Ling, S. (2022). Arousal-based pupil modulation is dictated by luminance. Scientific Reports, 12(1), 1390. https://doi.org/10.1038/s41598-022-05280-1

Reilly, J., Kelly, A., Kim, S. H., Jett, S., & Zuckerman, B. (2018). The human task-evoked pupillary response function is linear: Implications for baseline response scaling in pupillometry. Behavior Research Methods, 1–14.

Pg. 8, I do not know what ‘IEEE sentences’ are– consider writing out acronyms and providing accompanying references

Pg. 8, I do not know what ‘HINT sentences’ are – consider writing out acronyms and providing accompanying references

Packages such as 'GazeR' implement velocity-based blink detection and interpolation procedures with the idea that observations corresponding to partial eyelid closure are part of the blink. It is not clear how/why you implemented your particular choice of blink correction algorithms. Consider elaborating

It is not clear how you were able to obtain reliable pupil readings for the low (0 Lux) light condition. In our own past studies using luminance manipulations, we struggled to acquire reliable data in ‘pure’ dark even using an IR sensor with an Eyelink 1000. The eyetracker simply could not reliably discriminate pupil from iris in complete darkness. How did you shield the room from monitor luminance, etc. to achieve 0 Lux luminance?

6. PLOS authors have the option to publish the peer review history of their article (what does this mean?). If published, this will include your full peer review and any attached files.

Reviewer #1: No

Reviewer #2: **Yes: **Jamie Reilly

---

## [Author Response · Author response to Decision Letter 0]

19 Aug 2022

Reviewer #1: This manuscript describes a study to the effect of luminance and SNR on the peak pupil response dilation. Although the study methods are sound, the embedding of the study in the context of existing study can be improved substantially. Please find my suggestions below.

1. Embedding in existing literature. The authors indicate that the study described in the manuscript is one of few studies addressing the combined influence of luminance and (auditory) task load on the pupil response (e.g. line 562). However, the authors did not include a rigorous description of current work. Specifically, the authors should include the studies cited below (and probably more relevant studies can easily be found) to provide a more comprehensive overview of the current state of knowledge. The current studies up to date suggest that the current results are inconclusive, namely some studies have found evidence for an effect of luminance on the PPD evoked by cognitive tasks, and some did not show this effect. The authors could expand the discussion about the factors that may be associated with this current inconclusive evidence.

We thank our reviewers and academic editor to point us to all the relevant literature. We realized that many papers suggested are very helpful to improve the overall quality of our paper. We have included these studies in the introduction and discussion to improve the review of past studies (line88) and possible reasons that lead to inconsistencies in the existing studies (line114-122, line411 -430, line440).

2. Currently, most published “guidelines” on pupillometry recommend to use a medium illumination level. How would the current results affect this recommendation? Importantly, the authors report a main effect of luminance, and the post-hoc analysis shows that the pupil response in bright conditions differs from that in medium level conditions. However, the difference between the dark and medium level condition is not significant. Regardless, the remaining part of the manuscript treats this absent effect as if this difference was indeed significant (e.g. line 395). Please adapt (why would you test the main effect if you don’t follow up on the results appropriately?).

We agree with the reviewer that the main effect of luminance is not thoroughly followed up, therefore, we added extensive discussion on the results and why some results are different from past studies (i.e., medium luminance is bigger than both dark and bright) (line411-430). From our current results, in practice medium-luminance is still recommended (line591), and we added a note on how this might not be realistic in clinical settings with current setups and how this could be improved.

3. Related to the incomplete account of the current literature, the authors do not explain and discuss the potential physiological mechanisms that may account for (part of) the effects. Specifically, although Steinhauer’s work is cited, they do not discuss the notion that the pupil response in darkness might be affected by a limited effect of the PNS on the pupil size in dark. Please include a discussion on the separate roles of PNS and SNS in the pupil response (and the effect of illumination)

Indeed, we had not discussed the ANS mechanism sufficiently. We added a section in discussion to better explain/interpret our results (line411-440).

4. The authors suggest that luminance effects on the pupil response to cognitive load may be a negative aspect, and that it may “hinder” (e.g., line 82, line 120) the pupillometry applications. Rather, such effects can be exploited in case these are meaningful (e.g. Steinhauers work). Please discuss the potential benefits of such approaches.

Indeed, the complex relation between SNS and PNS can be experimentally controlled in a meaningful way (e.g., Steinhauer et al., 2004; Reilly et al., 2019) to answer interesting research questions. We have changed the tone when addressing the impact of luminance to be more neutral and added a note on how we can use the knowledge to benefit scientific research (line430). But we still keep and emphasize the central message of our paper that in the case of standardizing clinical pupillometry, factors like luminance and individual differences should be properly examined and controlled across clinics, in order to extract the pupillary responses to clinical stimuli of interest (i.e., standardized test words or sentences, pure tone or FM/AM tone for audiometry, visual stimuli etc) (line580). We think this message is more important (and also much-needed) for the field of cognitive hearing science. 

5. Line 68: a larger effect does not automatically indicate a predominant effect. Please rephrase.

Thanks, we have rephrased ‘predominant’ to ‘having a larger impact’

6. Line 136 and line 386: treating the baseline as being associated with the “tonic LC state” is probably not appropriate in the current study, as the baseline was derived just before the actual sentence presentation. The tonic LC state is observed during rest. The participants are not passively listening but probably actively anticipating the target speech. See also Joshi, S., & Gold, J. I. (2020). Pupil size as a window on neural substrates of cognition. Trends in cognitive sciences, 24(6), 466-480.

Absolutely, thank you for pointing this out to us. We have removed the tonic/phasic state, and clarified that our baseline is not resting state baseline (line472).

7. Please provide the rationale and hypotheses for the (additional) analyses performed. Please also provide the rationale for the specific analyses performed on the behavioural data. Are these analyses robust against the fact that the behavioural performances were at ceiling levels?

We have added a section head to better explain our rationale for performing additional analyses (line 451). We also pointed more often to the additional analyses in the method and discussion section to support the value of the additional analyses (line 265 line 274, line410, line581). We believe that this will make the additional analyses section more relevant and will tie better with our central message. Behavioural results were indeed close to ceiling, as intended, due to participants being normally hearing and relatively young. The logistic mixed effect model does take into account such upper asymptote in performance and address the contrast better than applying transformations like arcsine-square-root (line212). 

8. The sentence duration needs to be provided in the main methods section. How is the varying sentence length accounted for with respect to the interval in which the pupil data were analysed? Was the interval adapted relative to the length of the shortest sentence? The additional analyses do not support this.

The sentences remained unchanged and the analytical window contained sentence duration plus 2s of fixed waiting interval starting at the offset of the sentence for the pupil peak to emerge. We added further clarification to this point in line270. The uncorrected sentence duration is common in past listening effort studies because standardized sentences are almost always of slight duration difference (Winn et al., 2018, page 19). We also reported the 10th, 30th, 50th, 70th and 90th percentile of the distribution of all sentences (line 509). This distribution could give a better indication of the sentence duration. This duration variability does not affect averaged pupil response, as long as the time alignment of traces is done consistently (either by the onset or the offset of the sentence). We reported in the additional analyses the results of doing both types of alignment to ensure readers of our methods and to explore whether different alignment would change results significantly. A formal analysis was also performed to examine systematically whether variability in sentence duration, albeit small, has any impact on calculating task-evoked pupillary response.

9. In many studies, the dynamic range of the pupil size that is used in the proportional baseline correction is specifically and separately measured (not during the actual task). This differs from the current approach. Could this have affected the results?

We agree that resting state baseline would change the dynamic range calculation and baseline-corrected pupil response. And such baseline could also be affected by luminance. Considering the arousal that would take place when getting ready for a task, we further speculate that the resting state baseline would be lower than that we considered (prior to a given block, or across a whole experiment). In other words, the dynamic range we showed is presumably underestimated, compared to a method which would take the resting-state baseline instead. But note that other phenomena could also affect resting baseline other than task readiness. In our protocol, we did not include resting state pupil recordings, therefore, it is difficult for us to make any informed claim in this regard. We clarified that our baseline is not a resting state baseline and that we could get different results if we had done so. (line472). 

10. The resolution of the figures is poor, also when downloading the images directly from the portal.

We have addressed this issue and uploaded images of correct resolution.

11. The authors already interpret the data in the results section (e.g. lines 374-375). In addition, the authors could use more formal language throughout the paper (e.g. line 566: what is meant with “gripping”?).

We have removed the data interpretation in the results section, except for one short sentence where we wish to highlight the key finding of the paper (line384). Also, we have improved the formality of the language to avoid using ‘gripping’ or other ambiguous words.

12. Line 409: serperation typo

The typo is now corrected.

13. Lines 407-409: this is speculative; please indicate that this is not based on the current evidence.

We agree; it is indeed a speculation based on the shape we obtained from the three luminance levels tested. We added possible support from the past literature (line 440), but neither of them provides direct support for this speculation, due to different test materials (Pan et al., 2022) and physiological biomarkers used (Slade et al., 2021). We have rephrased the sentence to highlight that this was only a speculation and more studies are needed to consolidate it.

Reviewer #2: This well-written and interesting work highlights a question of methodological significance in pupillometry (i.e., how should we account for variable lighting conditions?). The study was conducted with admirable rigor, and the data will contribute to our evolving understanding of how to optimize pupillometry and control for the all-important factor of luminance. 

Thank you for your kind words. We’re glad you found it interesting. 

This reviewer had several questions that the authors might wish to consider prior to this work:

One of the central claims of this work is that little is known about the interaction between task difficulty and luminance that might inform best practices in baseline correction. I would argue on this point that this is not entirely correct. The authors have missed several studies examining the effects of luminance perturbations on the magnitude of the task-evoked pupillary response function (Pan et al., 2022; Reilly et al., 2018). In addition, there are other past studies uncited (e.g., Bradhshaw, 1969) that involved similar manipulations from a half century ago. It is reasonably well-accepted that linear (subtraction) is the most biologically plausible form of baseline pupil correction (as referenced by Mathot et al). It might be useful to make contact with some of the following sources on the effects of luminance:

Bradshaw, J. L. (1969). Background light intensity and the pupillary response in a reaction time task. Psychonomic Science, 14(6), 271 -272. https://doi.org/10.3758/BF03329118

Pan, J., Klímová, M., McGuire, J. T., & Ling, S. (2022). Arousal-based pupil modulation is dictated by luminance. Scientific Reports, 12(1), 1390. https://doi.org/10.1038/s41598-022-05280-1

Reilly, J., Kelly, A., Kim, S. H., Jett, S., & Zuckerman, B. (2018). The human task-evoked pupillary response function is linear: Implications for baseline response scaling in pupillometry. Behavior Research Methods, 1–14.

We are incredibly grateful for this insight and the many suggestions on the literature. We have integrated these papers in the introduction and discussion to improve the coverage of past studies (line88) and possible reasons that could have led to inconsistencies in the existing studies (line114-122, line411 -430, line440).

Pg. 8, I do not know what ‘IEEE sentences’ are– consider writing out acronyms and providing accompanying references

Pg. 8, I do not know what ‘HINT sentences’ are – consider writing out acronyms and providing accompanying references

Thanks for pointing out the acronym issue. We have added the corpus name and the reference to the materials.

Packages such as 'GazeR' implement velocity-based blink detection and interpolation procedures with the idea that observations corresponding to partial eyelid closure are part of the blink. It is not clear how/why you implemented your particular choice of blink correction algorithms. Consider elaborating

Thank you for making us aware of these packages. At the time of the experimentation and data analysis, GazeR was not published (Apr 2020) so it did not come to our attention. Instead, we followed pupillometry guidelines provided in Winn et al., 2008 and Mathôt et al., 2018 when constructing pupil pre-processing pipeline. We reported in the Methods section the detailed steps and corresponding Matlab and R functions for analysis to promote reproducibility. Note that we also integrated a gaze-based exclusion (line233) rule, different from GazeR. We believe this addition could improve data quality, especially in settings like ours where participants looked at a very big screen and gaze wander-offs were common (easier than they might have been in another study with a typical laptop screen). Removing pupil measures with gaze too far away from the fixation cross can effectively control for the inattentive moments. 

It is not clear how you were able to obtain reliable pupil readings for the low (0 Lux) light condition. In our own past studies using luminance manipulations, we struggled to acquire reliable data in ‘pure’ dark even using an IR sensor with an Eyelink 1000. The eyetracker simply could not reliably discriminate pupil from iris in complete darkness. How did you shield the room from monitor luminance, etc. to achieve 0 Lux luminance?

Most likely, this is because it wasn’t exactly 0 lux. Please have another read at the experimental room setup (line184), and you can find an actual photo of the setup below. The participants sat on a chair 2m away from a large screen (35-inch). In the dark condition, the room lights were turned off, and the screen luminance was set to the lowest for signaling participants. The room was indeed dark, but still with some dots of light emitted from the big screen. When we measured luminance using the luxometer TES-1335 at participants’ eye level 2m away from the screen, the reading was zero due to low screen light, big distance from the light source and possibly the sensitivity of the luxometer. But we acknowledge that we were not shielding all the light in the room, and agree with the reviewer that ‘complete darkness’ is not scientifically rigid. So, we have changed all occurrences of wording ‘complete darkness’ to ‘darkness’ or ‘close to 0lux’.

---

## [Decision Letter · Decision Letter 1]

22 Sep 2022

PONE-D-22-07841R1Luminance effects on pupil dilation in speech-in-noise recognitionPLOS ONE

Dear Dr. Zhang,

Thank you for submitting your revised manuscript to PLOS ONE. Both of the original reviewers read the revision and you will happy to see that they are almost satisfied with the revision, although they both do raise a few final comments. I invite you to respond to these comments in a second revision. I do not anticipate sending the manuscript out for review again, but of course I reserve the right to do so if for some reason I feel that this is necessary. A small practical request from my side: In the previous revision, you kept several layers of track changes, which I imagine reflects the back-and-forths between the various authors. However, this is really confusing for the reviewers and myself! Could you indicate the changes in a simpler way, for example a single layer of track changes or by highlighting the relevant parts of the text?

We look forward to receiving your revised manuscript.

Kind regards,

Sebastiaan Mathôt, Ph.D.

Academic Editor

PLOS ONE

Journal Requirements:

Reviewers' comments:

Reviewer's Responses to Questions

**Comments to the Author**

1. If the authors have adequately addressed your comments raised in a previous round of review and you feel that this manuscript is now acceptable for publication, you may indicate that here to bypass the “Comments to the Author” section, enter your conflict of interest statement in the “Confidential to Editor” section, and submit your "Accept" recommendation.

Reviewer #1: (No Response)

Reviewer #2: (No Response)

2. Is the manuscript technically sound, and do the data support the conclusions?

Reviewer #1: Yes

Reviewer #2: Yes

3. Has the statistical analysis been performed appropriately and rigorously? 

Reviewer #1: Yes

Reviewer #2: Yes

4. Have the authors made all data underlying the findings in their manuscript fully available?

Reviewer #1: Yes

Reviewer #2: No

5. Is the manuscript presented in an intelligible fashion and written in standard English?

Reviewer #1: No

Reviewer #2: Yes

6. Review Comments to the Author

Reviewer #1: Review of PONE-D-22-07841_R1 “Luminance effects on pupil dilation in speech-in-noise recognition”.

The authors have improved the readability of the manuscript and better acknowledge the current state of literature in the introduction and discussion sections. I still have a few comments and questions:

Readability: The manuscript still contains quite a few typos and non-optimal phrases. I list a couple of them below, but there are probably more. Please carefully check the quality of the English text. Examples: line 46 “bigger and bigger”; line 71: this sentence doesn’t read fluently; line 75: “dark lighting” is a bit confusing, line 84: “it”: not clear what it refers to; line 105: “staining”, line 136: “they” should be there, line 140: “on” is missing, line 236: “as” should be and, line 256: add “rate”, line 296: this line refers to a result that hasn’t been presented yet, which is confusing; line 337: “our end goal” is not a clear phrase; line 339: sentence is not fluent, line 346: “disregarded” does not fit in the sentence; line 352: “massive”: informal language; which also holds for “little larger” in the next line.

Language: The authors describe that two types of sentences were used, either IEEE or HINT, depending on the native tongue of the participant. Please describe the characteristics of these sentence sets in more detail, as (differences in) complexity, sentence structure, sentence length etc. can all affect the performance and pupil response. This is especially relevant in relation to the analysis of the effect of sentence duration (was sentence set confounding, i.e. was the distribution of sentence duration the same for both sets?) and the determination of the analysis window. The analysis would benefit from an explicit comparison of the effect of sentence set / language to check whether it affected the dependent measures (even though the limited sample size in such a comparison probably reduces the chances of observing an effect). Line 299 and line 528: Please explain what is meant with “with different speech materials”? Did the listeners perceive another (third) set of sentences? Please describe the details.

Line 311: Why do the authors refer to the current task as a “non-demanding” task? Please explain.

Figure 2: to me, it is not clear why the pupil response is quiet, which seems relatively large in Figure 2 (plot showing the response over time) is actually relatively small in the plot showing mean PPD. What is causing the relative difference between the size of the effect between these plots? How do the authors explain the relatively large pupil response in the most easy condition?

Absent interaction between SNR and luminance: none of the many analyses found an interaction effect between SNR and luminance. However, the wording of some sentences is a bit confusing and seems to refer to such an interaction. For example: line 392: “interact”: please rephrase.

Discussion: The way in which additional analyses and results are reported in the discussion section is not usual and not introduced properly. Consider moving these sections to the results section.

Reviewer #2: The authors have addressed many of this reviewer's original concerns but might consider the following issues:

1) Effect size measurements should appear in conjunction with relevant statistics to promote replication.

2) It is unclear why the authors chose to implement proportional baseline scaling. This technique is not recommended as a baseline correction procedure as it can significantly distort the magnitude of evoked pupil dilation at high/low levels of the dynamic range. For example, an evoked change of .1mm from a 1mm baseline is a much higher %change relative to the same absolute change (.1mm) from a 9mm baseline. Several recent papers have addressed this issue along with recommendations for subtractive baseline scaling.

3) It is reasonably well established that at very high ends of the dynamic range of the pupil (e.g., intense light, absolute darkness), pupillary movements become idiosyncratic. In addition, eye trackers often become unreliable in these conditions because of challenges in contrasting pupil from iris. It was not clear what the overarching recommendations were here regarding control for luminance. In moderate ambient lighting conditions as most labs might encounter (e.g., fluorescent lighting) practical variability would not be too high (assuming testing in a windowless room).

4) I had difficulty finding links to a data repository to examine stimuli, etc.

7. PLOS authors have the option to publish the peer review history of their article (what does this mean?). If published, this will include your full peer review and any attached files.

Reviewer #1: No

Reviewer #2: **Yes: **Jamie Reilly

---

## [Author Response · Author response to Decision Letter 1]

6 Nov 2022

We thank the two reviewers and the academic editor for their constructive comments which helped us improve this manuscript. We have made several edits to further polish this submission.

We apologize for the double edits in the earlier revision and fixed it in this new version to avoid confusion.

Note that line numbers are referenced to the clean version of this revision (not with tracked changes).

Reviewer #1: Review of PONE-D-22-07841_R1 “Luminance effects on pupil dilation in speech-in-noise recognition”.

The authors have improved the readability of the manuscript and better acknowledge the current state of literature in the introduction and discussion sections. I still have a few comments and questions:

Readability: The manuscript still contains quite a few typos and non-optimal phrases. I list a couple of them below, but there are probably more. Please carefully check the quality of the English text. Examples: line 46 “bigger and bigger”; line 71: this sentence doesn’t read fluently; line 75: “dark lighting” is a bit confusing, line 84: “it”: not clear what it refers to; line 105: “staining”, line 136: “they” should be there, line 140: “on” is missing, line 236: “as” should be and, line 256: add “rate”, line 296: this line refers to a result that hasn’t been presented yet, which is confusing; line 337: “our end goal” is not a clear phrase; line 339: sentence is not fluent, line 346: “disregarded” does not fit in the sentence; line 352: “massive”: informal language; which also holds for “little larger” in the next line.

We thank the reviewer to point out those non-optimal wordings and typos. We have made corresponding changes and also additional checks for typos (line 34, line 46, line71, line84, line105, line140, line 236, line256, line 296, line334, line337, line346,line352, line355 ).

Language: The authors describe that two types of sentences were used, either IEEE or HINT, depending on the native tongue of the participant. Please describe the characteristics of these sentence sets in more detail, as (differences in) complexity, sentence structure, sentence length etc. can all affect the performance and pupil response. This is especially relevant in relation to the analysis of the effect of sentence duration (was sentence set confounding, i.e. was the distribution of sentence duration the same for both sets?) and the determination of the analysis window. The analysis would benefit from an explicit comparison of the effect of sentence set / language to check whether it affected the dependent measures (even though the limited sample size in such a comparison probably reduces the chances of observing an effect). Line 299 and line 528: Please explain what is meant with “with different speech materials”? Did the listeners perceive another (third) set of sentences? Please describe the details.

We thank the reviewer for requesting the precision of the sentences used. We added clarification in the section of method and additional analysis that the sentence distribution calculation is done on all the sentences, and the analysis window is from the sentence onset to the repeat prompt (line 291, line511). We also clarified in line229 and line528 that the ‘different’ materials referred to sentence not used in the previous experiment.

We agree that the HINT and the IEEE materials are different, not only in their length but also their semantic complexity. The analysis presented in Additional analysis deals to some extent with this variable since effects of duration are pooled together across the 2 materials, so there was more representation of French sentences in the long and very-long bins and more representation of English sentences in the short and very-short bins. However, we do not wish to draw the reader's attention to this language-induced difference because it is not a factor of interest in our scientific hypothesis, and it is not clear that the effect of language applies differentially to French and English participants. In other words, French participants may be used to relatively longer sentences than English participants (because of the way each language is constructed). So, the effect of duration might apply over different ranges in each language. This speculation is something for future empirical studies to resolve. We do not have the statistical power in this study to examine it properly, considering the number of main effect factors we are already investigating. What we ensured in the experimental design is that each participant performed the task in their native language, hence no extra cognitive effort required for processing non-native language. In this regard, the language of the materials was irrelevant. None of the present findings are about cognitive load due to lack of mastery in the materials. Therefore, effects of duration (and necessarily materials) have been postponed to the very end of the discussion (last section of Additional analyses) to make sure that the reader focuses on the take-home message of the paper which is about light effect and their interaction with SNR.

Line 311: Why do the authors refer to the current task as a “non-demanding” task? Please explain.

Speech in noise tasks conducted between 0 and 14 dB SNR are generally not considered difficult for NH listeners. The speech intelligibility performance is high (>90%) and approaching ceiling (line 292). The wording of “non-demanding” is consistent with previous literature in NH listeners. 

Figure 2: to me, it is not clear why the pupil response in quiet, which seems relatively large in Figure 2 (plot showing the response over time) is actually relatively small in the plot showing mean PPD. What is causing the relative difference between the size of the effect between these plots? How do the authors explain the relatively large pupil response in the most easy condition?

This difference is due to the inherent variability in the individual pupil traces. When averaging traces who peak at slightly different times, the resulting average is smoothed down and could have a peak latency that is also slightly different from the average of the individual peak latencies. That is true of pupillometry just like many other neurophysiological techniques. For the PPDs shown in the errorbar plots and in statistical analysis, we calculate one PPD from aggregated 20 traces in a list, and then those more stereotypical traces are entered into statistics and calculating the Grand-average trace.This is similar to the difference when we calculate PPD based on single trial and based on list aggregation (PPD of a trace aggregated from 20 traces), in Supplementary materials.

In response to the latter question, the quiet condition did not differ significantly from +14 or even +7 dB; it’s really at 0 dB SNR that PPD amplitude and latency started to differ. So, we didn’t delve into this speculation. 

Absent interaction between SNR and luminance: none of the many analyses found an interaction effect between SNR and luminance. However, the wording of some sentences is a bit confusing and seems to refer to such an interaction. For example: line 392: “interact”: please rephrase.

This is essentially because we suspect the interaction to exist but could not demonstrate it. For sake of rigor, we clarified (start of the second paragraph of the discussion of Exp2) that: “The lack of significant interaction between luminance and SNR conditions in our results show that this bias is relatively consistent across SNRs”. However, in the rest of this paragraph, we did delve into the speculation that an interaction might exist given a more global understanding of the pattern in our data as well as the existing literature. 

Discussion: The way in which additional analyses and results are reported in the discussion section is not usual and not introduced properly. Consider moving these sections to the results section.

We have considered the re-organisation. However, the current format best suits the logic of the paper. The result section directly reflects our answers to the initial hypothesis. Then upon checking the results, further analyses were required in order to better understand possible reasons to explain our rather surprising results, i.e., we did not expect the effect of luminance to be of such a big impact on task-evoked response. To move the additional analysis to the results section might confuse the readers with the central message of our article and the flow of scientific hypothesis testing. 

Reviewer #2: The authors have addressed many of this reviewer's original concerns but might consider the following issues:

1) Effect size measurements should appear in conjunction with relevant statistics to promote replication (mean diff, pooled std of the groups).

We thank the reviewer for pointing this out. We have added in the results section group mean and std where relevant (line289, line325, line 384, line391, line425)

2) It is unclear why the authors chose to implement proportional baseline scaling. This technique is not recommended as a baseline correction procedure as it can significantly distort the magnitude of evoked pupil dilation at high/low levels of the dynamic range. For example, an evoked change of .1mm from a 1mm baseline is a much higher %change relative to the same absolute change (.1mm) from a 9mm baseline. Several recent papers have addressed this issue along with recommendations for subtractive baseline scaling.

We did not choose proportional baseline scaling but a subtractive method. This was exactly the point of one section in Additional Analysis Effect of PPD calculation methods. In the main results reporting section (line263), we used subtractive method exactly due to the reasons proposed by the reviewer.

3) It is reasonably well established that at very high ends of the dynamic range of the pupil (e.g., intense light, absolute darkness), pupillary movements become idiosyncratic. In addition, eye trackers often become unreliable in these conditions because of challenges in contrasting pupil from iris. It was not clear what the overarching recommendations were here regarding control for luminance. In moderate ambient lighting conditions as most labs might encounter (e.g., fluorescent lighting) practical variability would not be too high (assuming testing in a windowless room).

We agree with the reviewer that in summary we recommend moderate ambient lighting for pupillometry experiments (line596). Our paper serves as a scientific investigation of how in low and high light, the effect of light is beyond just physiological constraints, but also with the task-evoked pupillary response. Previous studies acknowledged in the introduction had suggested that the relation between task-evoked and light-evoked pupillary response is complex and not completely consistent. Therefore, our study further contributes to this line of study to investigate this relation between two types of pupillary responses. We also made further analysis to explain the confounds of other possible factors in the experimental design that were not done in previous studies. Altogether, we believe that we presented a rigorous scientific case that luminance control is very important in pupillometry experiments and should be emphasized more in methods reports and experimental design.

4) I had difficulty finding links to a data repository to examine stimuli, etc.

To ease the access to data, we provided a zip file of all behavioural and pupil raw data with the submission. The zip file should appear at the end of the manuscript visible to the reviewer. However, we do not have the copyright to share the sentence stimuli (IEEE and HINT sentences) and they are only used in the experiment after the authorization of the owner and distributor of the materials.

---

## [Editor Report · Decision Letter 2]

18 Nov 2022

Luminance effects on pupil dilation in speech-in-noise recognition

PONE-D-22-07841R2

Dear Dr. Zhang,

Thank for submitting the revision of your manuscript. I did not send it out for review again, but rather checked myself whether all issues were addressed. And it is my pleasure to inform you that yes they have, and so your manuscript is herewith accepted for publication! I did still notice a handful of typos. My suggestion would be to wait until you receive the proofs, and then correct these. Thank you for contributing a valuable manuscript!

Kind regards,

Sebastiaan Mathôt, Ph.D.

Academic Editor

PLOS ONE
---

## [Editor Report · Acceptance letter]

25 Nov 2022

PONE-D-22-07841R2 

Luminance effects on pupil dilation in
speech-in-noise recognition 

Dear Dr. Zhang:

I'm pleased to inform you that your manuscript has been deemed suitable for publication in PLOS ONE. Congratulations! Your manuscript is now with our production department. 

Kind regards, 

on behalf of

Dr. Sebastiaan Mathôt 

Academic Editor

PLOS ONE